# Effect of high-flow nasal cannula therapy on mechanical ventilation duration in the pediatric intensive care unit

Jaeyoung Choi[1], Esther Park[2], Hyejeong Park[3,4], Danbee Kang[3,4], Jeong Hoon Yang[1], Hyunsoo Kim[3,4], Juhee Cho[3,4], Joongbum Cho[1] *

1 Department of Critical Care Medicine, Samsung Medical Center, Sungkyunkwan University School of Medicine, Seoul, Republic of Korea, 2 Department of Pediatrics, Jeonbuk National University Children's Hospital, Jeonju, Republic of Korea, 3 Center for Clinical Epidemiology, Samsung Medical Center, Seoul, Republic of Korea, 4 Department of Clinical Research Design & Evaluation, SAIHST, Sungkyunkwan University, Seoul, Republic of Korea

* joongbum.cho@gmail.com

## Abstract

### Background

High-flow nasal cannula (HFNC) therapy has gained popularity in the pediatric intensive care unit (PICU). However, the nationwide effect of HFNC on mechanical ventilation duration has not been studied.

### Methods

We retrospectively analyzed pediatric patients (28 days to 17 years old) admitted to tertiary ICUs for respiratory support from 2012 to 2019 using the Korean National Health Insurance database. Pre-/post-HFNC periods were defined as the 12 months before and after the application of HFNC in any hospital, respectively, allowing a 6-month transition period. Mechanical ventilation duration and ventilator-free days during these two periods were compared using a multivariable regression model.

### Results

Using data from 46 hospitals, 4,705 and 4,864 respective pre-/post-HFNC period patients were evaluated. During the post-HFNC period, 14.8% of patients were treated by HFNC, and 67.1% were treated using invasive mechanical ventilation. In adjusted analysis, mechanical ventilation duration was reduced by 0.99 days (confidence interval [CI]: -1.86, -0.12). The duration was significantly reduced by 17.81 days (CI: -35.46, -0.16) among patients whose ventilation duration was longer than 28 days. In subgroup analysis, mechanical ventilation duration was reduced by 1.49 days (CI: -2.78, -0.19) in the overall surgical group and 6.71 days (CI: -11.71, - 1.71) in the neurologic subgroup. Ventilator-free days were increased only in the overall surgical group, by 0.31 days (CI: 0.01, 0.61).

**Data Availability Statement:** This study utilized data obtained from the Korean Health Insurance Review & Assessment Service (HIRA). Due to legal

restrictions, the authors are not permitted to publicly share or distribute the dataset. However, researchers may request access to the HIRA database via the official HIRA Open Data Service portal (https://opendata.hira.or.kr/home.do) after obtaining the necessary approvals. Data access requests should specify the same parameters (time period, data terms, and claim codes) used in this study to facilitate comparable analysis.

**Funding:** The author(s) received no specific funding for this work.

**Competing interests:** The authors have declared that no competing interests exist.

**Abbreviations:** ICU, intensive care unit; HFNC, high-flow nasal cannula; VFDs, ventilator-free days; KNHI, Korean National Health Insurance..

## Conclusions

Application of HFNC to PICU patients could reduce mechanical ventilation duration, especially in patients requiring prolonged mechanical ventilator support or in post-operative patients.

## Introduction

For intubated patients in the intensive care unit (ICU), mechanical ventilation provides essential support for respiratory failure [1]. Previous pediatric studies have estimated that 41 to 55% of patients admitted to the ICU require invasive mechanical intubation [2–4]. However, prolonged duration of invasive mechanical ventilation increases resource burden and risk of complications, including ventilator-associated pneumonia and ICU-acquired weakness, which negatively affect outcomes [5, 6]. Therefore, effective measures to reduce the duration of mechanical ventilation are required to improve patient care.

High flow nasal cannula (HFNC) has gained popularity as a non-invasive respiratory support modality among ICUs due to its simplicity and high tolerability [7, 8]. While evidence suggested higher treatment failure rate compared to non-invasive positive pressure ventilation in children with acute respiratory failure, HFNC remained as an alternative way of respiratory support by lowering rate of treatment failure and elevation in respiratory care compared to standard oxygen [9–11]. The widespread use of HFNC is further supported by clinical efficacy demonstrated in diverse clinical settings including acute hypoxic respiratory failure, acute bronchiolitis, asthma, or obstructive apnea in the pediatric population [12–14]. Evidence also suggested that HFNC could help reduce reintubation rate in critically ill children and children after cardiac surgery [15, 16]. However, there is a lack of real-world national-level evidence regarding the efficacy of HFNC in ICU patients in general, and few studies have evaluated the effect of HFNC use on the duration of mechanical ventilation in pediatric patients.

This study therefore aimed to evaluate the effect of HFNC use in pediatric ICU (PICU)s on mechanical ventilation duration based on analysis of a nationwide database. The primary outcome of this study was to compare mechanical ventilation duration and ventilator-free days before and after the introduction of HFNC, and secondary outcome focused on subgroup analysis according to surgical status and diagnostic subgroups.

## Methods

### Study population and design

This was a population-based retrospective cohort study based on the Health Insurance Review and Assessment database of the Korean Ministry of Health. The vast majority (97%) of Koreans are covered by The Korean National Health Insurance (KNHI), while the remaining 3% of Koreans who cannot afford national insurance are covered by the Medical Aid Program. As the Health Insurance Review and Assessment service reviews reimbursement claims from KNHI and the Medical Aid Program, virtually all inpatient and outpatient visits, procedures, and prescriptions are available in the Health Insurance Review and Assessment database.

To evaluate the duration of mechanical ventilation and ventilation-free days (VFDs) before and after introduction of HFNC, tertiary hospitals were selected if they had issued at least one prescription for respiratory therapy including oxygen therapy (KNHI procedure codes: M0040, HFNC: M0046) or mechanical ventilation (KNHI procedure codes: M5850, M5857,

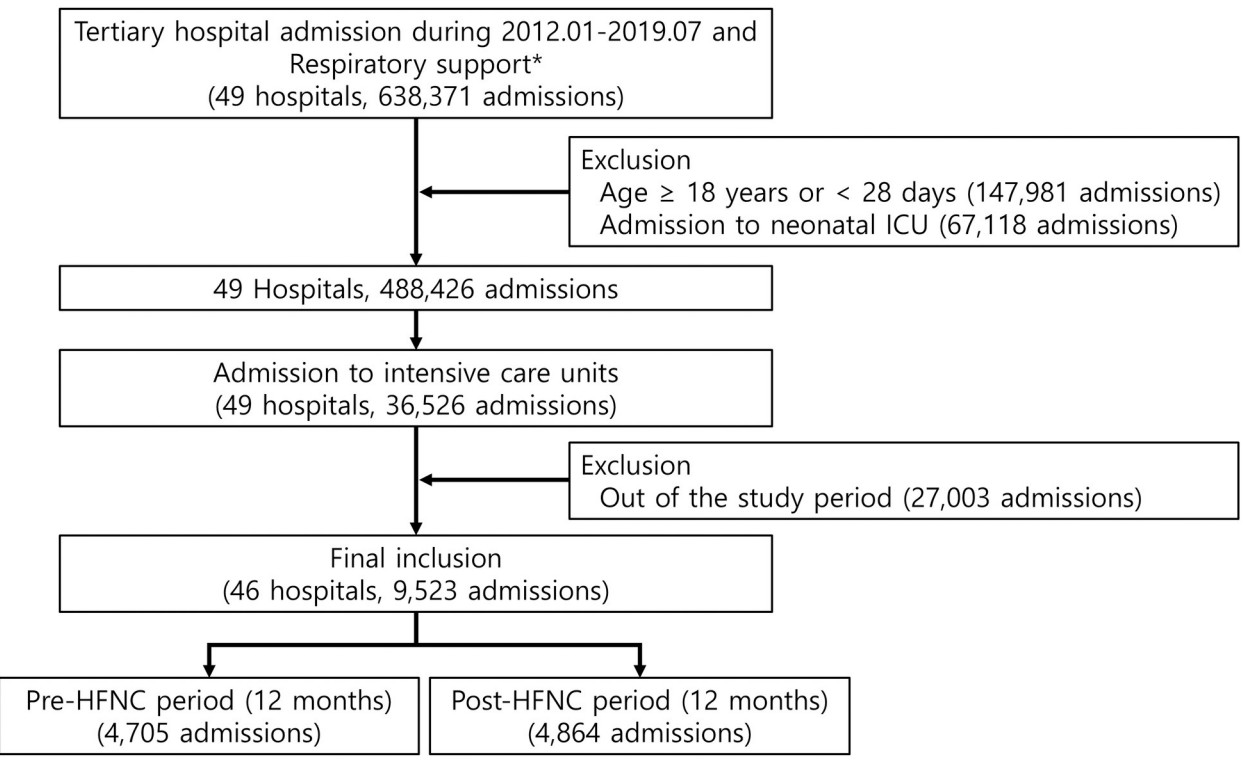

**Fig 1. Flow diagram of patient selection and exclusion.**

M5858, M5860) between 2012 and 2019 (638,371 admissions from 49 tertiary hospitals). For selected hospitals, pediatric populations including patients 17 years or younger were included. Neonates (<28 days of age) (N = 460,209) and neonatal ICU admissions (N = 67,188) were excluded due to lack of availability of individual identification (Fig 1).

The study was reviewed by the Institutional Review Board of Samsung Medical Center (IRB protocol 2019-12-156) and was exempted from the requirement for informed consent due to sole use of previously collected de-identified administrative data. The de-identified data can be accessed upon approval of Health Insurance Review and Assessment service, Korea. This study was conducted in accordance with the Strengthening the Reporting of Observational Studies in Epidemiology (STROBE) Statement, the guidelines for reporting observational studies.

### Measurements

**Pre- and post-HFNC periods.**   Pre-and post-HFNC application periods were defined to evaluate the effect of HFNC on duration of mechanical ventilation and VFDs in each hospital. While the first day of HFNC application at each hospital was set as the index date, the pre-HFNC period, transition period, and post-HFNC periods were defined as the previous 12 months from the index date and 6 months and 12 months after the index date, respectively.

**Duration of mechanical ventilation.**   The primary outcomes were mechanical ventilation duration and VFDs. Total usage and number of days for each code were used to calculate mechanical ventilation duration. For patients who did not need mechanical ventilation, mechanical ventilation duration was set as 0. VFDs states were defined as follows: (1) VFDs = 0 if the subject died within 28 days of mechanical ventilation; (2) VFDs = up to 28 days of mechanical ventilation; (3) VFDs = 0 if the subject was mechanically ventilated for >28 days.

**Other variables.** Demographic characteristics, hospital information, procedures, and prescriptions were obtained using KNHI claim codes. Classifications of congenital anomalies, neoplasms, neurologic diseases, respiratory diseases, circulatory diseases, injuries, gastrointestinal disease, and infectious diseases were obtained from claims data and defined using ICD-10 codes assigned upon hospitalization. Procedures of interest were mechanical ventilation (KNHI procedure codes: M5857, M5858, and M5860) and extracorporeal membrane oxygenation (procedure codes: O1901-O1904). Use of vasopressor drugs such as dobutamine, dopamine, epinephrine, and norepinephrine for >2 days was determined by Korean Drug and Anatomical Therapeutic Chemical codes.

## Statistical analysis

Mean and standard deviation or median and interquartile range were used to describe continuous variables. Chi-square test and Student's $t$ test were used to compare categorical and continuous variables, respectively.

To compare mechanical ventilation duration and VFDs before and after HFNC use, we calculated odds ratio (ORs) with 95% CIs using univariable and multivariable random-effect logistic regression analyses considering correlations arising within the same hospital over time. To compensate for potential confounding factors, we further adjusted for age, gender, primary diagnosis, region, admission department, and procedures/prescriptions during hospital admission (vasopressor drugs, extracorporeal membrane oxygenation, and hemodialysis). Subgroup analysis was conducted according to surgery; chest surgery; and diagnosis of congenital, neurologic, respiratory, circulatory, or infectious diseases at admission. To assess the linear relationship between change in VFDs and HFNC ratio in each hospital, regression analysis was conducted.

A p-value <0.05 was considered to be significant for all analyses. Statistical analyses were performed using SAS and Ⓡ Visual Analytics.

## Results

Between January 2012 and July 2019, the use of HFNC gradually increased after its introduction in 2015 (S1 Fig). In the 46 tertiary hospitals evaluated, 4,705; 2,301; and 4,864 patients received respiratory support within the 12-month period before HFNC introduction, the transition period, and the 12-month period after HFNC introduction, respectively (Fig 1).

In the post-HFNC period, patients were more likely to be younger (5.4 years vs. 5.2 years, p = 0.02), diagnosed with a congenital anomaly (36.5% vs. 38.4%, p = 0.001), and admitted to a hospital in a rural region (15.9% vs. 21.0%, p<0.001). Larger numbers of post-HFNC patients received mechanical ventilation support (64.8 vs. 67.1, p = 0.02) and required vasopressors (63.4 vs. 68.2%, p<0.01) than in the pre-HFNC period (Table 1). S1 Table describes the number and percentage of patients with mechanical ventilation in each subgroup.

In the adjusted model (Table 2), there was significant reduction in mechanical ventilation duration of 0.99 days in the post-HFNC period (95% CI -1.86, -0.12, p = 0.03), but differences between VFDs pre- and post-HFNC were not significant (coefficient = 0.1, 95% CI -0.17, 0.37, P value = 0.45).

When assessed by subgroup (Table 3), the adjusted duration of mechanical ventilation was significantly reduced by -1.49 days (CI: -2.78, -0.16), -1.47 days (CI: -2.54, -0.40), and -6.71 days (CI: -10.40, -0.34) in mechanically ventilated patients (> 0 days), surgical patients, and those with a neurologic diagnosis, respectively. In subgroup analysis of VFDs (Table 4), overall surgical group patients had a significant improvement in VFDs of 0.31 days.

**Table 1. Baseline characteristics of study patients.**

| | Overall (N = 9,569) | Pre-HFNC period (N = 4,705) | Post-HFNC period (N = 4,864) | p-value |
|---|---|---|---|---|
| **Age (years), median (IQR)** | 5.3 (5.5) | 5.4 (5.6) | 5.2 (5.4) | 0.02 |
| **Age group** | | | | 0.05 |
| Infants (<1) | 3,190 (33.3) | 1,528 (32.5) | 1,662 (34.2) | |
| Children (1–11) | 4,446 (46.5) | 2,183 (46.4) | 2,263 (46.5) | |
| Adolescents (12–17) | 1,933 (20.2) | 994 (21.1) | 939 (19.3) | |
| **Sex, male** | 5,451 (57.0) | 2,680 (57.0) | 2,771 (57.0) | 0.99 |
| **Primary diagnosis** | | | | 0.001 |
| Congenital anomaly | 3,588 (37.5) | 1,719 (36.5) | 1,869 (38.4) | |
| Neoplasm | 1,323 (13.8) | 669 (14.2) | 654 (13.4) | |
| Neurologic disease | 990 (10.3) | 471 (10.0) | 519 (10.7) | |
| Respiratory | 901 (9.4) | 421 (8.9) | 480 (9.9) | |
| Circulatory disease | 842 (8.8) | 461 (9.8) | 381 (7.8) | |
| Injury | 586 (6.1) | 300 (6.4) | 286 (5.9) | |
| Gastrointestinal disease | 269 (2.8) | 124 (2.6) | 145 (3.0) | |
| Not elsewhere classified | 255 (2.7) | 121 (2.6) | 134 (2.8) | |
| Infectious disease | 146 (1.5) | 62 (1.3) | 84 (1.7) | |
| Others | 669 (7.0) | 357 (7.6) | 312 (6.4) | |
| **Region** | | | | < .001 |
| Seoul | 6,227 (65.1) | 3154 (67.0) | 3,073 (63.2) | |
| Metropolitan | 1,573 (16.4) | 804 (17.1) | 769 (15.8) | |
| Rural | 1,769 (18.5) | 747 (15.9) | 1,022 (21.0) | |
| **Admission department** | | | | 0.22 |
| Medical | 4,600 (48.1) | 2,232 (47.4) | 2,368 (48.7) | |
| Surgical | 4,969 (51.9) | 2,473 (52.6) | 2,496 (51.3) | |
| **Surgery** | | | | 0.2 |
| No | 2,437 (25.5) | 1,171 (24.9) | 1,266 (26.0) | |
| Yes | 7,132 (74.5) | 3,534 (75.1) | 3,598 (74.0) | |
| **Intervention for critical care** | | | | |
| Vasopressor | 6,298 (65.8) | 2,982 (63.4) | 3,316 (68.2) | < .001 |
| ECMO | 154 (1.6) | 81 (1.7) | 73 (1.5) | 0.39 |
| Hemodialysis | 410 (4.3) | 199 (4.2) | 211 (4.3) | 0.79 |
| **Respiratory therapy** | | | | |
| Oxygen | 8,351 (87.3) | 4,128 (87.7) | 4,223 (86.8) | 0.18 |
| HFNC | 718 (7.5) | - | 718 (14.8) | - |
| Non-invasive mechanical ventilation | 170 (1.8) | 72 (1.5) | 98 (2.0) | 0.07 |
| Mechanical ventilation | 6,310 (65.9) | 3,047 (64.8) | 3,263 (67.1) | 0.02 |
| **Outcomes** | | | | |
| Hospital LOS (day), median (IQR) | 13 (9–24) | 13 (9–25) | 13 (9–23) | 0.10 |
| ICU LOS (day), median (IQR) | 3 (1–6) | 2 (1–6) | 3 (1–7) | 0.54 |
| Hospital mortality, n (%) | 522 (5.5) | 261 (5.5) | 261 (5.4) | 0.70 |

Values are presented as n (%) or median (IQR)

HFNC, high flow nasal cannula; ICU, intensive care units; LOS, length of stay; IQR, interquartile range; ECMO, extracorporeal membrane oxygenation

There was no association between the rate of HFNC usage and change in mean mechanical ventilation duration in individual hospitals (p-value = 0.13, correlation coefficient = 0.23, S2 Fig).

**Table 2. Mechanical ventilation duration and ventilation-free days according to period of HFNC use.**

| | Pre-HFNC period (N = 4,705) | Post-HFNC period (N = 4,864) | Crude model | | | Adjusted model* | | |
|---|---|---|---|---|---|---|---|---|
| | Median (IQR) | Median (IQR) | Coefficient | 95% CI | p-value | Coefficient | 95% CI | p-value |
| | Mean (SD) | Mean (SD) | | | | | | |
| Mechanical ventilation duration** | 1 (0–3) | 1 (0–4) | -0.76 | -1.65, 0.13 | 0.1 | **-0.99** | **-1.86, -0.12** | **0.03** |
| | 5.6 (27.6) | 4.8 (7.8) | | | | | | |
| Ventilation-free days at day 28 | 27 (24–28) | 27 (24–28) | -0.08 | -0.40, 0.23 | 0.61 | 0.1 | -0.17, 0.37 | 0.45 |
| | 23.6 (7.8) | 23.5 (7.8) | | | | | | |

HFNC, high flow nasal cannula; CI, confidence interval; IQR, interquartile range

Reference is the pre-HFNC period

*Model was adjusted for age, sex, primary diagnosis, region, admission department, vasopressor drugs, extracorporeal membrane oxygenation, and hemodialysis

**Patients who did not receive mechanical ventilation were assigned a value of zero

## Discussion

In this real-world database study, we evaluated the nationwide effect of HFNC application in the PICU on mechanical ventilation duration. We found a reduction in mechanical ventilation duration especially among prolonged mechanically ventilated patients and surgically treated patients. However, there was no reduction in the number of VFDs in a 28-day period. We also did not find a dose-response relationship between HFNC use and reduction in mechanical ventilation duration.

The mean mechanical ventilation duration was reduced by 0.8 days (from 5.6 to 4.8 days) after the transition period (Table 2). When we compared only mechanically ventilated patients, there was a reduction in ventilation duration of 1.4 days (from 8.6 to 7.2 days) after the transition period (S2 Table). Though a reduction of 0.8 days might seem small, it correlates

**Table 3. Mechanical ventilation duration according to HFNC period by subgroup.**

| MV duration by subgroup | Pre-HFNC period (N = 4,705) | Post-HFNC period (N = 4,864) | Crude model | | | Adjusted model* | | |
|---|---|---|---|---|---|---|---|---|
| | Median (IQR) | Median (IQR) | Coefficient | 95% CI | p-value | Coefficient | 95% CI | p-value |
| **MV status** | | | | | | | | |
| No MV or MV ≤ 28 days (n = 8,840) | 1 (0–3) | 1 (0–3) | 0.02 | -0.17, 0.20 | 0.87 | -0.10 | -0.27, 0.07 | 0.26 |
| MV >28 days (n = 729) | 51 (36–90) | 46 (35–69) | **-16.94** | **-33.88, -0.003** | **0.05** | **-17.81** | **-35.46. -0.16** | **0.05** |
| MV used (> 0 days, n = 6,310) | 2 (1–6) | 2 (1–6) | **-1.48** | **-2.80, -0.15** | **0.03** | **-1.49** | **-2.78, -0.19** | **0.02** |
| Surgical status | | | | | | | | |
| Overall surgical group (n = 5,163) | 2 (0–3) | 1 (0–3) | -1.09 | -2.19, 0.004 | 0.05 | **-1.47** | **-2.54, -0.40** | **0.007** |
| Chest surgery group (n = 1,615) | 2 (1–3) | 2 (1–2) | -0.14 | -0.86, 0.59 | 0.71 | -0.28 | -0.90, 0.35 | 0.38 |
| Diagnostic subgroup | | | | | | | | |
| Neurologic disease (n = 590) | 1 (0–5) | 1 (0–6) | **-5.37** | **-10.40, -0.34** | **0.04** | **-6.71** | **-11.71, -1.71** | **0.009** |
| Respiratory disease (n = 590) | 2 (0–6) | 3 (0–8) | 1.02 | -2.20, 4.25 | 0.53 | 0.65 | -2.54, 3.85 | 0.69 |
| Circulatory disease (n = 588) | 1 (0–2) | 1 (0–4) | -2.66 | -6.54, 1.23 | 0.18 | -3.37 | -7.16, 0.43 | 0.08 |

HFNC, high flow nasal cannula; CI, confidence interval; MV, mechanical ventilation

Reference is pre-HFNC period

*Model was adjusted for age, sex, primary diagnosis, region, admission department, vasopressor drugs, extracorporeal membrane oxygenation, and hemodialysis

**Table 4. Coefficient of ventilation-free days at day 28 according to period of HFNC use by subgroup.**

| MV duration by subgroup | Pre-HFNC period (N = 4,705) | Post-HFNC period (N = 4,864) | Crude model | | | Adjusted model* | | |
|---|---|---|---|---|---|---|---|---|
| | Median (IQR) | Median (IQR) | Coefficient | 95% CI | p-value | Coefficient | 95% CI | p-value |
| **MV status** | | | | | | | | |
| MV used (> 0 days, n = 6,310) | 26 (21–27) | 26 (21–27) | 0.13 | -0.29, 0.55 | 0.55 | 0.15 | -0.21, 0.51 | 0.42 |
| Surgical status | | | | | | | | |
| Overall surgical group (n = 5,163) | 27 (25–28) | 26 (25–28) | 0.03 | -0.31, 0.36 | 0.87 | **0.31** | **0.01, 0.61** | **0.04** |
| Chest surgery group (n = 1,615) | 26 (25–27) | 26 (26–27) | 0.14 | -0.21, 0.49 | 0.44 | 0.25 | -0.03, 0.53 | 0.08 |
| Diagnostic subgroup | | | | | | | | |
| Neurologic disease (n = 590) | 27 (22–28) | 27 (21–28) | -0.70 | -1.86, 0.45 | 0.23 | 0.33 | -0.68, 1.35 | 0.52 |
| Respiratory disease (n = 590) | 26 (22–28) | 25 (20–28) | -0.39 | -1.38, 0.60 | 0.44 | -0.19 | -1.09, 0.72 | 0.69 |
| Circulatory disease (n = 588) | 27 (25–28) | 27 (22–28) | -0.81 | -2.06, 0.44 | 0.2 | -0.51 | -1.57, 0.56 | 0.35 |

HFNC, high flow nasal cannula; CI, confidence interval; MV, mechanical ventilation

Reference is pre-HFNC period

*Model was adjusted for age, sex, primary diagnosis, region, admission department, vasopressor drugs, extracorporeal membrane oxygenation, and hemodialysis

** Patients who did not receive mechanical ventilation were assigned a value of zero

to a decrease of about 3800 (= 0.8 x 4800) ventilation-days per year nationwide. A mean mechanical ventilation period of 7 days corresponds to 543 (= 3800/7) fewer ventilated patients. This number indicates that HFNC significantly improves the efficacy of mechanical ventilation. The standard deviation of mechanical ventilation duration was larger in the pre-HFNC period (27.6) than in the post-HFNC period (7.8) (Table 2). Since the distribution of ventilation duration was skewed right, we speculate that the reduction in standard deviation was caused by narrowing of the right side of the distribution (right-deviated values corresponding to prolonged ventilation). Therefore, by reducing high utilization of mechanical ventilation by a small number of patients, HFNC can redistribute the utilization of mechanical ventilation in an efficient way. In the subgroup that required prolonged (>28 days) mechanical ventilation, median and mean ventilation durations were reduced by 5 days and 25 days, respectively (Table 2) (S2 Table). In contrast, in the subgroup that did not undergo mechanical ventilation or underwent ventilation for ≤ 28 days, there were no changes in mean or median ventilation days. This suggests that the effect of HFNC on reducing mechanical ventilation duration is more prominent in patients requiring prolonged (> 28 days) mechanical ventilation.

Compared to previous studies, this study had less selection bias because we included the entire pediatric population of Korea who required respiratory support at a tertiary hospital. Unexpectedly, the characteristics of the pre/post-HFNC period groups differed (Table 1). There was a 5% increase in rural region admissions, a 4.8% increase in vasopressor use, and a 2.3% increase in mechanical ventilation use among patients with respiratory support during the post-HFNC period. Nationally, the application of HFNC increased rapidly over several years (after initiation of KNHI coverage in 2015). To exclude the effect of the prolonged study time, we individualized the transition period between the pre/post-HFNC periods according to each hospital's initiation of HNFC. However, this might not have adequately addressed the systemic bias induced by the long study period and corresponding medical environment changes. Another possibility is that these changes were induced by HFNC application. One possible reason for the increase in rural admissions is that the introduction of HFNC reduced transport to urban hospitals. As the technical skills to set up HFNC are relatively easy to learn [17], rural hospitals would have been able to provide additional respiratory support other than

conventional oxygen therapy during the post-HFNC period. However, HFNC could also potentially increase inter-hospital transport if used during transport by reducing the requirement for invasive ventilation during transport [18]. Increased mechanical ventilation during the post-HFNC period was also an unexpected finding. This paradoxical increase has also been observed in some previous bronchiolitis studies. A global increase in ICU admissions and rates of mechanical ventilation were observed in pediatric patients with bronchiolitis, and an association with HFNC was suggested as a possible cause [19–22]. One hypothesis is that HFNC protocols can introduce novel criteria for ICU transfer or intubation (a certain oxygen concentration or flow rate) [23]. The reason underlying the observed increase in vasopressors needs to be investigated further. When we adjusted for the different characteristics pre- and post-HFNC, the reduction in mechanical ventilation duration (0.99 days) was similar to the unadjusted mean reduction. However, there was no reduction in the mean and standard deviation of VFDs or the median and interquartile range (Table 2). Considering the similar hospital mortality (5.5 vs. 5.4%, Table 1) and 28-day mortality during mechanical ventilation (4.5 vs. 4.5%, S3 Table), prolonged mechanical ventilation (> 28 days) in the pre-HFNC period might have contributed to the discrepancy between mechanical ventilation duration and VFDs (S4 Table).

Previous studies have supported a reduction in mechanical ventilation duration with the use of HFNC among adult COVID-19 patients [24]. HFNC is also related to lower extubation failure and reintubation rates, possibly contributing to a reduction in mechanical ventilation duration [25, 26]. Although more pediatric studies are required, one retrospective single-center study demonstrated reduction in intubation rate and ventilation duration after introduction of HFNC in a PICU [27]. The effect of HFNC on mortality rate is unclear. While one study warned that HFNC increased mortality by inappropriately delaying intubation, other meta-analyses and randomized controlled studies have found no effects of HFNC on mortality [12, 28, 29].

In subgroup analyses, the surgical subgroup showed a reduction in mechanical ventilation duration (1.47 days) and VFDs (0.31 days) in the adjusted model (Tables 3 and 4). A previous meta-analysis reported that prophylactic HFNC use in the immediate postoperative period after cardiothoracic surgery reduced the reintubation rate compared with conventional oxygen therapy [30]. The authors suggested that improved respiration of patients with obesity or a pre-existing respiratory disease contributed to this finding. A pediatric randomized study showed that the preventive HFNC group had a better lung ultrasound score and less atelectasis than the conventional oxygen group in children ($\leq$ 2 years) receiving general anesthesia (>2 hours). End expiratory pressure generated by HFNC was suggested as one of the mechanisms [31]. In our study, the chest surgery group did not show a significant reduction in mechanical ventilation duration or VFD in the adjusted model. These different results might have been due to application of preventive or prophylactic HFNC in previous studies compared to the pragmatic approach used in our study.

In the neurologic subgroup, a significant decline (6.7 days) in mechanical ventilation duration was observed in the adjusted model (Table 3). A previous study reported decreases in post-extubation respiratory failure and reintubation rates with use of HFNC in neurocritical patients [32]. HFNC might help maintain airway patency among neurologic patients, thereby reducing the reintubation rate [32–34]. However, the differences in mechanical ventilation duration in this subgroup could also have arisen due to differences in mortality in this subgroup; this changed from 3.4% in the pre-HFNC period to 5.2% in pre-HFNC period (S3 Table). Possible explanations for the mortality change include the increased use of HFNC in palliative care, including brain death, rather than the direct effects of HFNC. As noted in previous studies, HFNC can be used in end-of-life situations to comfort patients and to avoid invasive support [35–37].

This study had several limitations. First, data collected in the national database are primarily collected for administrative purposes, and the lack of physiologic data in this database hampered our ability to evaluate HFNC use according to respiratory severity. However, we adjusted for severity using patient characteristics, hospital factors, and treatment requirements such as extracorporeal membrane oxygenation, hemodialysis, and vasopressors. Second, the use of a historic control group exposes the study to systemic bias between the two periods. To address this, we individualized the transition period focusing on the initiation of HFNC at each hospital to reduce systematic errors. Third, we could not distinguish whether the use of HFNC was before or after mechanical ventilation, but in either case, the advantages of reduced mechanical ventilation duration remained the same. Fourth, the result may not be reproducible under different medical environments. In Korea, the use of non-invasive positive pressure ventilation was limited due to the lack of interface during the study period. 58 non-invasive ventilators were available in 18 tertiary hospitals in 2015 [38]. The main strength of this study lies in its nationwide scale; inclusion of all tertiary hospitals in Korea without selection bias enabled the evaluation of the real-world effects of HFNC in PICU patients.

Our study findings have several clinical implications. Use of HFNC in the ICU may reduce mechanical ventilation duration, notably in post-surgical patients and patients that require prolonged mechanical ventilation. The lack of decrease in VFDs with HFNC use, however, indicates the need for further studies to determine specific indications for HFNC use in the ICU, especially in acute periods.

In conclusion, the application of HFNC in PICU patients may reduce mechanical ventilation duration, especially in patients requiring prolonged mechanical ventilator support or post-operative patients.

## Supporting information

**S1 Fig. Annual description of respiratory support.** MV, mechanical ventilation; HFNC, high-flow nasal cannula therapy.
(TIF)

**S2 Fig. Use rate of high-flow nasal cannula therapy as a respiratory support in the intensive care unit of each hospital (X-axis) and changes in mean mechanical ventilation duration (Y-axis).** HFNC rate: rate of HFNC use among all respiratory support modalities (oxygen therapy, high flow nasal cannula, mechanical ventilation). Mechanical ventilation difference (days): mean ventilation duration of the post-HFNC period minus that of the pre-HFNC period in each hospital (negative value indicates a reduction in ventilation duration after HFNC application). HFNC, high-flow nasal cannula therapy.
(TIF)

**S1 Table. Number and percentage of patients with mechanical ventilation by subgroups.**
(DOCX)

**S2 Table. Mean and standard deviation of mechanical ventilation duration according to HFNC period.**
(DOCX)

**S3 Table. Number of patients who died within 28 days of receiving mechanical ventilation.**
(DOCX)

**S4 Table. Mean and standard deviation of ventilation-free days at day 28 according to HFNC period.**
(DOCX)

## Author Contributions

**Conceptualization:** Jaeyoung Choi, Esther Park, Danbee Kang, Jeong Hoon Yang, Juhee Cho, Joongbum Cho.

**Data curation:** Hyejeong Park, Danbee Kang, Jeong Hoon Yang, Juhee Cho.

**Formal analysis:** Hyejeong Park, Danbee Kang, Hyunsoo Kim, Juhee Cho.

**Investigation:** Jaeyoung Choi.

**Supervision:** Danbee Kang, Jeong Hoon Yang, Juhee Cho, Joongbum Cho.

**Visualization:** Hyejeong Park.

**Writing – original draft:** Jaeyoung Choi, Esther Park, Hyejeong Park.

**Writing – review & editing:** Jaeyoung Choi, Esther Park, Danbee Kang, Jeong Hoon Yang, Hyunsoo Kim, Juhee Cho, Joongbum Cho.

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
