## [Decision Letter · Decision Letter 0]

20 Sep 2024

PONE-D-24-30283Effect of high-flow nasal cannula therapy on mechanical ventilation duration in the pediatric intensive care unitPLOS ONE

Dear Dr. Cho,

Thank you for submitting your manuscript to PLOS ONE. After careful consideration, we feel that it has merit but does not fully meet PLOS ONE’s publication criteria as it currently stands. Therefore, we invite you to submit a revised version of the manuscript that addresses the points raised during the review process.

**ACADEMIC EDITOR:**It is necessary to look at the statistical part and assess which statistical methods would need to be performed, given the inequality of the sample distribution.Revise statistical part according to the second reviewer's comments.==============================

We look forward to receiving your revised manuscript.

Kind regards,

Stefan Grosek, Ph.D., M.D.,

Academic Editor

PLOS ONE

Journal requirements: 1. When submitting your revision, we need you to address these additional requirements. Please ensure that your manuscript meets PLOS ONE's style requirements, including those for file naming. The PLOS ONE style templates can be found at https://journals.plos.org/plosone/s/file?id=wjVg/PLOSOne_formatting_sample_main_body.pdf and https://journals.plos.org/plosone/s/file?id=ba62/PLOSOne_formatting_sample_title_authors_affiliations.pdf. 2. We note that you have indicated that there are restrictions to data sharing for this study. For studies involving human research participant data or other sensitive data, we encourage authors to share de-identified or anonymized data. However, when data cannot be publicly shared for ethical reasons, we allow authors to make their data sets available upon request. For information on unacceptable data access restrictions, please see http://journals.plos.org/plosone/s/data-availability#loc-unacceptable-data-access-restrictions.  Before we proceed with your manuscript, please address the following prompts: a) If there are ethical or legal restrictions on sharing a de-identified data set, please explain them in detail (e.g., data contain potentially identifying or sensitive patient information, data are owned by a third-party organization, etc.) and who has imposed them (e.g., a Research Ethics Committee or Institutional Review Board, etc.). Please also provide contact information for a data access committee, ethics committee, or other institutional body to which data requests may be sent. b) If there are no restrictions, please upload the minimal anonymized data set necessary to replicate your study findings to a stable, public repository and provide us with the relevant URLs, DOIs, or accession numbers. Please see http://www.bmj.com/content/340/bmj.c181.long for guidelines on how to de-identify and prepare clinical data for publication. For a list of recommended repositories, please see https://journals.plos.org/plosone/s/recommended-repositories. You also have the option of uploading the data as Supporting Information files, but we would recommend depositing data directly to a data repository if possible. Please update your Data Availability statement in the submission form accordingly.

Additional Editor Comment:

Dear Authors

We received two excellent reviews, where the first reviewer practically has no significant reservations about the article itself, how it is designed and presented, while the second reviewer, although he is satisfied with the article as a whole, believes that it is necessary to look at the statistical part and assess which statistical methods would need to be performed, given the inequality of the sample distribution.

So please look especially to the statistical part of your study and answer to the reviewers in the best way you can.

Kind regards

Reviewers' comments:

Reviewer's Responses to Questions

**Comments to the Author**

1. Is the manuscript technically sound, and do the data support the conclusions?

Reviewer #1: Yes

Reviewer #2: Partly

2. Has the statistical analysis been performed appropriately and rigorously? 

Reviewer #1: Yes

Reviewer #2: No

3. Have the authors made all data underlying the findings in their manuscript fully available?

Reviewer #1: Yes

Reviewer #2: Yes

4. Is the manuscript presented in an intelligible fashion and written in standard English?

Reviewer #1: Yes

Reviewer #2: Yes

5. Review Comments to the Author

Reviewer #1: Review of Manuscript PONE- D-24-30283

Authors, overall:

• Overall, excellent work and the recommendations ae minor and are simply to enhance your efforts.

• My recommendation is acceptance of your manuscript with consideration of the recommendations below to enhance the excellent quality of your work.

• A more detailed review of each of the above areas is outlined below.

Abstract

• Excellent overall description of background introduction, methods, results and conclusion.

Introduction

• Very good description of background issues and goals of study.

• Recommendations to Authors:

• Describe pediatric ICU as PICU as commonly seen in other studies to satisfy readers intuition.

• Describe other accepted means of HFNC use in pediatrics to include

acute hypoxemic respiratory failure, post-extubation, post-operative patients to include cardio-thoracic patients, obstructive lung disease such as asthma and bronchiolitis.

• Discuss prior work comparing efficacy and limitations of HFNC to standard oxygen therapy (SOT), nasal CPAP (n-CPAP) and NIPPV (Bi-PAP).

• Describe mechanical ventilation duration and ventilator-free days as the primary outcomes. Describe sub-group analysis of length of mechanical ventilation, surgical status and diagnostic subgroups as secondary outcomes.

Methods

• Excellent database source and description.

• Excellent inclusion criteria and exclusion of NICU patients in presence of known age and physiologic related differences in addition to age-related co-morbidities.

• Good description of VFD and “other variables”.

• Good description of intended primary outcomes.

• Appropriate statistical analysis.

• Recommendations to authors: none.

Results

• Good overall description would be enhanced by the following.

• Recommendations to authors:

• Figure 1 is not true flow diagram or reflective of patient selection and exclusion; rename as “annual description of respiratory support” and provide “p-values” to outline statistically significant difference between both eras (pre- and post- HFNC initiation).

• Table 1 – rename as baseline characteristics of study patients and categorize hospital, PICU length of stays and mortality as “Outcomes”.

• Paragraph 3 – list “Table 2” earlier…”In the adjusted model (Table 2) there was significant reduction” and not at the end of the paragraph so that readers can more easily refer to the table and note your accurate description of the result.

• Paragraph 4- list “Table 4” earlier…”When assessed by subgroup (Table 4), the adjusted duration…” and not at the end of the paragraph so that readers can more easily refer to the table and note your accurate description of the result.

• Page 10 - provide statistical data (p-values, correlation co-efficient) to support your likely correct contention that there was no association between rate of HFNC usage and change in mean mechanical ventilation duration in individual hospitals and briefly add the same information to Figure 2 to provide the same statistical absence of association.

Tables, Figures

• Page 26 – line 465 should read S2 Table, not S1 Table

• Page 26 – line 466 should read S1 Table, not S2 Table.

Discussion:

• ↑ use of HFNC since initiation 2015.

• Baseline characteristics - younger age, anomalies, rural location, vasopressor use.

• Key findings:

Increase use of mechanical ventilation.

Duration ↓if ≥ 28 days and in.

post-operative surgical and neurologic diagnosis.

No decrease in VFD in general but was present in surgical patients.

Implications:

• HFNC is effective in post-operative patients and those with prolonged mechanical ventilation (≥ 28 days).

• Facilitates availability of PICU beds in tertiary care centers.

• Facilitates transfer of those patients requiring higher level of care through avoidance of complications and risks of invasive mechanical ventilation during transport process.

• Absence of reduction in VFD is likely due to pragmatic use of HFNC in this study compared to prophylactic use in other or future studies.

Future work:

• Evaluate effectiveness of HFNC to increase VFD when used empirically to avoid mechanical ventilation and to earlier successful extubation.

• Multi-center, international study to evaluate generalizability regarding effective use of HFNC to facilitate reduction in duration of mechanical ventilation, availability of tertiary PICU beds through use at other urban community hospitals and rural locations.

Reviewer #2: This retrospective cohort study looks at a national database and aims to capture all of the children managed on PICU for periods before and after the introduction of High Flow Nasal Cannula oxygen therapy on to each unit over a period of 7 years from 2012-2019. The authors have not commented on whether they have followed the STROBE guidelines, although from reading the paper, most of these have been achieved. The authors should clarify if the STROBE guidelines have been followed.

The authors note that between the 2 periods across the units studied there were differences in the characteristics of the groups, with the later period having more younger children, more children with congenital anomalies, more admissions to rural hospitals and a possibly sicker population with greater use of vasopressors. This may have influenced some of the findings and the authors do comment on this in the discussion.

The statistics as described in the methods appear appropriate, however there are issues when these are used in the results. Firstly the authors found that there was a difference in the duration of ventilation overall, but no statistically significant difference in ventilator free days at 28 days between the "before" and "after" groups. This is consistent with the issue that most children were post-operative (almost 75%) and only ventilated for a very short time (median ventilation duration for both groups 1 day). Also only a small proportion of children were supported with High Flow Nasal Cannula oxygen (14.8% in the "after" group) so its use had not become ubiquitous. Also as the authors note it is unclear from their source data when the High Flow Nasal Cannula oxygen was applied either pre-ventilation or following invasive ventilation. It is also not clear if some children were supported with High Flow Nasal Cannula oxygen without receiving invasive ventilation, which is another limitation for this study.

The biggest issue for the results is that means are used to obtain statistically significant results from distributions that are significantly skewed and do not follow a normal distribution. In these instances, the mean is not an appropriate calculation and only medians should be used. Given that the main effect noted by the authors appeared to be in children ventilated for more than 28 days, this is especially the case. This should also lead to more caution in the conclusion such that "may" should replace "can" in the last sentence of the discussion. There is an association but it cannot be concluded to be causal as there is too little other information. Likewise a reduction of 0.31 days in the length of ventilation (i.e. less than 8 hours) probably has little clinical significance.

The authors have also not commented on other aspects of Paediatric Intensive Care that might be affected by High Flow Nasal Cannula oxygen such as Length of Stay. One of the benefits of this support reported elsewhere is that it can be used outside of Critical Care areas and so may hasten PICU discharge. Also the use of High Flow Nasal Cannula oxygen did not seem to affect the use of mechanical ventilation in terms of the percentage of patients invasively ventilated. Understanding the percentages invasively ventilated across each patient type e.g. surgical, respiratory etc, would also be useful.

In terms of the particular populations and practice of Paediatric Intensive Care in South Korea, the authors should reflect that there patient cohort is somewhat different to those described elsewhere within national databases, with a far higher percentage of post-operative children admitted compared to Western nations reporting similar data. They should also comment on the fact that non-invasive support, which again is widely used particularly in Europe, and often employed to either avoid invasive ventilation or to support children post-extubation, was almost non-existent in South Korean PICUs. High Flow Nasal Cannula oxygen support is often seen as an alternative to Non-Invasive Ventilation in many countries, but clearly this was not the case in South Korea, and the authors should note that in this paper.

A retrospective cohort study such as this does have limitations and the authors do appear to have recognised some of these in their discussion, but this should be expanded somewhat to included some of the issues noted above.

In the tables, especially Tables 3 and 4, it would be appropriate to include the number of children in each row, to provide an indication of the numbers being reported. Whilst Table 1 does provide some of this data, it would still be better to repeat some of these figures to give greater clarity.

Finally in terms of the figures, just one scale should be used for the Y-axis, given that the percentages are so close, and especially given that the changes in High Flow Nasal Cannula use tend towards the right side of the figure but that Y-axis percentage applies to the percentage overall on oxygen. Figure 2 also adds very little and should be omitted. The slope of the line is so minimal and looks somewhat arbitrary given the overall distribution.

6. PLOS authors have the option to publish the peer review history of their article (what does this mean?). If published, this will include your full peer review and any attached files.

Reviewer #1: No

Reviewer #2: **Yes: **Peter J Davis

---

## [Author Response · Author response to Decision Letter 0]

4 Nov 2024

Since it is not allowed to inserte tables and figures in this box, we have attached the 'Response to Reviewers' along with the manuscript to include responses to each reviewers' point and changes made in tables and figures. Please check the attached 'Response to Reviewers' file along with our response below to confirm the changes made in the tables and figures.

ACADEMIC EDITOR:

• It is necessary to look at the statistical part and assess which statistical methods would need to be performed, given the inequality of the sample distribution.

• Revise statistical part according to the second reviewer's comments.

Thank you for your comment. We agree with the academic editor’s comments and the reviewers’ comments that inequal distribution of our study prompted a careful selection of the statistical method. We have reviewed the statistical methods of our study and concluded that the use of the statistical method was not inappropriate, first due to the large number, and second due to the fact that the assumption of normality and homogeneity of variance is not necessary in logistic regression analysis. However, we agree the second reviewer’s comments and made following changes:

1) Have answered the second reviewer’s comment regarding the inclusion of mostly post-operative patients, the non-ubiquitous use of HFNC, and the uncertainty of the timing of HFNC in association with mechanical ventilation as follows:

Thank you for the comment. We recognize the limitation of our study regarding the association of the use of HFNC and the use of mechanical ventilation. We also agree with the comment that the use of HFNC was not ubiquitous. However, we have shown in supporting figure 2 that the rate of HFNC use upto 50% did not correlate with the mechanical ventilation duration. Therefore, we believe that the effect of HFNC on the mechanical ventilation duration remains the same regardless of the ubiquitous use of HFNC. Furthermore, since the definition of ventilator-free days includes those who have not been mechanically ventilated as ventilator-free days of 0, the comparison of ventilator-free days before and after HFNC initiation allowed us to compare patients regardless of the use of mechanical ventilators.

2) Changed the conclusion as the second reviewer’s comment with following response:

Thank you for the comment. We agree with the importance of median in our data as it was skewed right. However, we believe that if we mention the direction of skewness in data, it would be more informative to readers to know both mean and median. We also agree that caution be taken in conclusion and therefore have amended the last sentence of the discussion as follows:

(Line 340-342) “In conclusion, the application of HFNC in PICU patients may reduce mechanical ventilation duration, especially in patients requiring prolonged mechanical ventilator support or post-operative patients.”

3) Added supporting table 1 and revised table 3 and table 4 to clarify the number and the percentage of mechanically ventilated patients in each subgroups.

4) Revised the figure 2 as supporting figure 2 and omitted the line slope.

5) Revised the manuscript to point out the difference of medical environment in Korea compared to western countries and include the limitation of retrospective nature of the study .

(Line 324-331) “Second, use of a historic control group exposes the study to systemic bias between the two periods. To address this, we individualized the transition period focusing on the initiation of HFNC at each hospital to reduce systematic errors. Third, we could not distinguish whether the use of HFNC was before or after mechanical ventilation, but in either case, the advantages of reduced mechanical ventilation duration remained the same. Fourth, the result may not be reproducible under different medical environment. In Korea, the use of non-invasive positive pressure ventilation was limited due to the lack of infrastructure during the study period. 58 non-invasive ventilators were available in 18 tertiary hospitals in 2015”

Reviewer #1: Review of Manuscript PONE- D-24-30283

Authors, overall: 

• Overall, excellent work and the recommendations ae minor and are simply to enhance your efforts.

• My recommendation is acceptance of your manuscript with consideration of the recommendations below to enhance the excellent quality of your work.

• A more detailed review of each of the above areas is outlined below.

Abstract

• Excellent overall description of background introduction, methods, results and conclusion.

Introduction

• Very good description of background issues and goals of study. 

• Recommendations to Authors:

• Describe pediatric ICU as PICU as commonly seen in other studies to satisfy readers intuition.

Thank you for your comment. We have changed the phrase ‘pediatric ICU’ to ‘PICU’ as follows:

(Line 109-110) “This study therefore aimed to evaluate the effect of HFNC use in pediatric ICU (PICU)s on mechanical ventilation duration and ventilator-free days based on analysis of a nationwide database.”

• Describe other accepted means of HFNC use in pediatrics to include 

acute hypoxemic respiratory failure, post-extubation, post-operative patients to include cardio-thoracic patients, obstructive lung disease such as asthma and bronchiolitis. 

Thank you for your comment. We have revised the manuscript to encompass indications of HFNC as suggested. 

(Line 101-105) “The widespread use of HFNC is further supported by clinical efficacy demonstrated in diverse clinical settings including acute hypoxic respiratory failure, acute bronchiolitis, asthma, or obstructive apnea in the pediatric population [12-14]. Evidence also suggested that HFNC could help reduce reintubation rate in critically ill children and children after cardiac surgery.”

• Discuss prior work comparing efficacy and limitations of HFNC to standard oxygen therapy (SOT), nasal CPAP (n-CPAP) and NIPPV (Bi-PAP). 

Thank you for your comment. We have supplemented the introduction to include current evidence of HFNC compared to non-invasive ventilation and standard oxygen therapy.

 (Line 97-101) “While evidence suggested higher treatment failure rate compared to non-invasive positive pressure ventilation in children with acute respiratory failure, HFNC remained as an alternative way of respiratory support by lowering rate of treatment failure and elevation in respiratory care compared to standard oxygen”

• Describe mechanical ventilation duration and ventilator-free days as the primary outcomes. Describe sub-group analysis of length of mechanical ventilation, surgical status and diagnostic subgroups as secondary outcomes.

Thank you for your comment. We have revised the manuscript to clarify the primary and secondary outcomes. 

 (Line 110-113) “The primary outcome of this study was to compare mechanical ventilation duration and ventilator-free days before and after the introduction of HFNC, and secondary outcome focused on subgroup analysis according to surgical status and diagnostic subgroups.”

Methods

• Excellent database source and description.

• Excellent inclusion criteria and exclusion of NICU patients in presence of known age and physiologic related differences in addition to age-related co-morbidities.

• Good description of VFD and “other variables”.

• Good description of intended primary outcomes.

• Appropriate statistical analysis. 

• Recommendations to authors: none.

Results

• Good overall description would be enhanced by the following.

• Recommendations to authors: 

• Figure 1 is not true flow diagram or reflective of patient selection and exclusion; rename as “annual description of respiratory support” and provide “p-values” to outline statistically significant difference between both eras (pre- and post- HFNC initiation).

Thank you for your comment. We sincerely apologize for the wrong location of figure 1. We have mistakenly uploaded the other version of supporting figure 1 as figure 1. We present the correct version of figure 1. In addition, we have changed the name of the supporting figure 1 as “Annual description of respiratory support”. Also, we have found out that, due to the large number of cases included, p-values comparing the oxygen therapy and mechanical ventilation between the pre-and post- HFNC initiation eras were below 0.01. while the rate of oxygen therapy and mechanical ventilation were similar (oxygen therapy rate of 89.4% in pre-HFNC and 87.0% in post-HFNC era, and mechanical ventilation rate of 63.2% in pre-HFNC and 65.6% in post-HFNC era). This indicates that difference is statistically significant but clinically not significant. Also, since the initiation of HFNC varied in each hospital, we worried that directly comparing the rate of oxygen therapy and mechanical ventilation before and after 2015 (the year of HFNC initiation in Korea) would not accurately represent the effect of HFNC. We also worried it would cause confusion to readers about our point that the use of mechanical ventilation and oxygen therapy showed no significant change after the initiation of HFNC. Therefore, with all due respect to the reviewer’s comment, we decided not to present the p-values.

• Table 1 – rename as baseline characteristics of study patients and categorize hospital, PICU length of stays and mortality as “Outcomes”. 

Thank you for your comment. We have changed the name of Table 1 and categorized hospital length of stay, PICU length of stay and mortality as outcomes:

• Paragraph 3 – list “Table 2” earlier…”In the adjusted model (Table 2) there was significant reduction” and not at the end of the paragraph so that readers can more easily refer to the table and note your accurate description of the result.

Thank you for your comment. We have repositioned the phase “Table 2” accordingly to improve reader’s understanding.

(Line 195-196) “In the adjusted model (Table 2), there was significant reduction in mechanical ventilation duration of 0.99 days in the post-HFNC period (95% CI -1.86, -0.12, P value = 0.03),”

• Paragraph 4- list “Table 4” earlier…”When assessed by subgroup (Table 4), the adjusted duration…” and not at the end of the paragraph so that readers can more easily refer to the table and note your accurate description of the result.

Thank you for your comment. We have moved the phase “Table 3” and “Table 4” accordingly to reduce reader’s inconvenience.

(Line 199-200) “When assessed by subgroup (Table 3), the adjusted duration of mechanical ventilation was significantly reduced”

(Line 202-203) “In subgroup analysis of VFDs (Table 4), overall surgical group patients had a significant improvement in VFDs of 0.31 days.”

• Page 10 - provide statistical data (p-values, correlation co-efficient) to support your likely correct contention that there was no association between rate of HFNC usage and change in mean mechanical ventilation duration in individual hospitals and briefly add the same information to Figure 2 to provide the same statistical absence of association.

Thank you for your comment. We have revised the manuscript and figure 2 accordingly to provide p-values. However, we renamed the figure 2 as supporting figure 2 and removed the trend line according to the other reviewer’s comment.

(Line 204-206) “There was no association between the rate of HFNC usage and change in mean mechanical ventilation duration in individual hospitals (p-value=0.13, correlation coefficient=0.23, S2 Fig).”

Tables, Figures

• Page 26 – line 465 should read S2 Table, not S1 Table

Thank you for your comment. We give our sincere apology for malpresentation of the S1 table and S2 table. We have corrected the previous S1 Table and S2 Table. Due to the additional supporting table as S1 Table, previous S1 Table became S2 Table.

 (Line 492-3) “S2 Table. Mean and standard deviation of mechanical ventilation duration according to HFNC period.”

• Page 26 – line 466 should read S1 Table, not S2 Table.

Thank you for your comment. We give our sincere apology for malpresentation of the S1 table and S2 table. We have corrected the previous S1 Table and S2 Table. Due to the additional supporting table as S1 Table, previous S2 Table became S3 Table.

 (Line 494) “S3 Table. Number of patients who died within 28 days of receiving mechanical ventilation.”

Discussion:

• ↑ use of HFNC since initiation 2015.

• Baseline characteristics - younger age, anomalies, rural location, vasopressor use.

• Key findings: 

 Increase use of mechanical ventilation.

 Duration ↓if ≥ 28 days and in.

 post-operative surgical and neurologic diagnosis.

 No decrease in VFD in general but was present in surgical patients. 

Implications:

• HFNC is effective in post-operative patients and those with prolonged mechanical ventilation (≥ 28 days).

• Facilitates availability of PICU beds in tertiary care centers.

• Facilitates transfer of those patients requiring higher level of care through avoidance of complications and risks of invasive mechanical ventilation during transport process.

• Absence of reduction in VFD is likely due to pragmatic use of HFNC in this study compared to prophylactic use in other or future studies.

Future work: 

• Evaluate effectiveness of HFNC to increase VFD when used empirically to avoid mechanical ventilation and to earlier successful extubation. 

• Multi-center, international study to evaluate generalizability regarding effective use of HFNC to facilitate reduction in duration of mechanical ventilation, availability of tertiary PICU beds through use at other urban community hospitals and rural locations.

Confidential Comments to Editor 

Professor Grosek,

I have reviewed PLOS ONE manuscript PONE-D-24-30283. 

I recommend that it be accepted for publication with minor revisions as presented to authors. Overall, the conception, design, objectives, methods, presented results, conclusions and implications are well presented and the revisions are minor to enhance the author’s excellent work.

I appreciate the opportunity to review this manuscript and remain available through my email address (JLaham3@gmail.com) for any further questions or issues.

Best regards 

James Laham, D.O. 

Reviewer #2: This retrospective cohort study looks at a national database and aims to capture all of the children managed on PICU for periods before and after the introduction of High Flow Nasal Cannula oxygen therapy on to each unit over a period of 7 years from 2012-2019. The authors have not commented on whether they have followed the STROBE guidelines, although from reading the paper, most of these have been achieved. The authors should clarify if the STROBE guidelines have been followed.

Thank you for your comment. We have revised the method section of the manuscript to clarify that we have followed the STROBE guidelines.

(Line 137-139) “This study was conducted in accordance with the Strengthening the Reporting of Observational Studies in Epidemiology (STROBE) Statement, the guidelines for reporting observational studies.”

The authors note that between the 2 periods across the units studied there were differences in the characteristics of the groups, with the later period having more younger children, more children with congenital anomalies, more admissions to rural hospitals and a possibly sicker population with greater use of vasopressors. This may have influenced some of the findings and the authors do comment on this in the discussion.

The statistics as described in the methods appear appropriate, however there are issues when these are used in the results. Firstly the authors found that there was a difference in the duration of ventilation overall, but no statistically significant difference in ventilator free days at 28 days between the "before" and "after" groups. This is consistent with the issue that most children were post-operative (almost 75%) and only ventilated for a very short time (median ventilation duration for both groups 1 day). Also only a small proportion of children were supported with Hi

---

## [Decision Letter · Decision Letter 1]

2 Dec 2024

Effect of high-flow nasal cannula therapy on mechanical ventilation duration in the pediatric intensive care unit

PONE-D-24-30283R1

Dear Dr. Cho,

We’re pleased to inform you that your manuscript has been judged scientifically suitable for publication and will be formally accepted for publication once it meets all outstanding technical requirements.

Kind regards,

Stefan Grosek, Ph.D., M.D.,

Academic Editor

PLOS ONE

Additional Editor Comments (optional):

Dear Authors

Thanks for the corrections suggested by the reviewers. The article is now better and clearer. I will suggest to the editor that it be accepted for publication.

Kind regards

Academic Editor

Reviewers' comments:

Reviewer's Responses to Questions

**Comments to the Author**

1. If the authors have adequately addressed your comments raised in a previous round of review and you feel that this manuscript is now acceptable for publication, you may indicate that here to bypass the “Comments to the Author” section, enter your conflict of interest statement in the “Confidential to Editor” section, and submit your "Accept" recommendation.

Reviewer #1: All comments have been addressed

Reviewer #2: All comments have been addressed

2. Is the manuscript technically sound, and do the data support the conclusions?

Reviewer #1: Yes

Reviewer #2: Yes

3. Has the statistical analysis been performed appropriately and rigorously? 

Reviewer #1: Yes

Reviewer #2: Yes

4. Have the authors made all data underlying the findings in their manuscript fully available?

Reviewer #1: Yes

Reviewer #2: Yes

5. Is the manuscript presented in an intelligible fashion and written in standard English?

Reviewer #1: Yes

Reviewer #2: Yes

6. Review Comments to the Author

Reviewer #1: (No Response)

Reviewer #2: Thank you for taking on board the various comments and observations related to the initial draft of this paper. It is now clearer that the proposed benefits of High Flow Nasal Cannula oxygen, in terms of reduced ventilation durations, were primarily seen in children who were ventilated for the longest periods (>28 days) and in surgical patients. You have also appropriately noted that there are limitations when using this type of national administrative data in a cohort study, and that the findings may not be replicated in PICUs outside of Korea, given the specifics of care within tour country.

7. PLOS authors have the option to publish the peer review history of their article (what does this mean?). If published, this will include your full peer review and any attached files.

Reviewer #1: **Yes: **James L Laham, D.O., FAAP

Reviewer #2: **Yes: **Dr Peter J Davis

---

## [Editor Report · Acceptance letter]

4 Dec 2024

PONE-D-24-30283R1 

PLOS ONE

Dear Dr. Cho, 

I'm pleased to inform you that your manuscript has been deemed suitable for publication in PLOS ONE. Congratulations! Your manuscript is now being handed over to our production team.

Kind regards, 

on behalf of

Professor Stefan Grosek 

Academic Editor

PLOS ONE